# Role of Stem Cell-Derived Exosomes and microRNAs in Spinal Cord Injury

**DOI:** 10.3390/ijms241813849

**Published:** 2023-09-08

**Authors:** Jinsu Hwang, Sujeong Jang, Choonghyo Kim, Sungjoon Lee, Han-Seong Jeong

**Affiliations:** 1Department of Physiology, Chonnam National University Medical School, Hwasun 58128, Republic of Korea; wlstn0128@naver.com (J.H.); sujeongjang@jnu.ac.kr (S.J.); 2Department of Neurosurgery, Kangwon National University School of Medicine, Chuncheon 24341, Republic of Korea; jeuelkim@gmail.com; 3Department of Neurosurgery, Samsung Medical Center, Sungkyunkwan University School of Medicine, Seoul 06351, Republic of Korea; potata98@gmail.com

**Keywords:** spinal cord injury, mesenchymal stem cell, extracellular vesicles, exosome, microRNA

## Abstract

Neurological disorders represent a global health problem. Current pharmacological treatments often lead to short-term symptomatic relief but have dose-dependent side effects, such as inducing orthostatic arterial hypotension due to the blockade of alpha receptors, cardiotoxic effects due to impaired repolarization, and atrioventricular block and tachycardia, including ventricular fibrillation. These challenges have driven the medical community to seek effective treatments for this serious global health threat. Mesenchymal stem cells (MSCs) are pluripotent cells with anti-inflammatory, anti-apoptotic, and immunomodulatory properties, providing a promising alternative due to their ability to differentiate, favorable culture conditions, in vitro manipulation ability, and robust properties. Although MSCs themselves rarely differentiate into neurons at the site of injury after transplantation in vivo, paracrine factors secreted by MSCs can create environmental conditions for cell-to-cell communication and have shown therapeutic effects. Recent studies have shown that the pleiotropic effects of MSCs, particularly their immunomodulatory potential, can be attributed primarily to these paracrine factors. Exosomes derived from MSCs are known to play an important role in these effects. Many studies have evaluated the potential of exosome-based therapies for the treatment of various neurological diseases. In addition to exosomes, various miRNAs derived from MSCs have been identified to regulate genes and alleviate neuropathological changes in neurodegenerative diseases. This review explores the burgeoning field of exosome-based therapies, focusing on the effects of MSC-derived exosomes and exosomal miRNAs, and summarizes recent findings that shed light on the potential of exosomes in the treatment of neurological disorders. The insights gained from this review may pave the way for innovative and effective treatments for these complex conditions. Furthermore, we suggest the therapeutic effects of exosomes and exosomal miRNAs from MSCs, which have a rescue potential in spinal cord injury via diverse signaling pathways.

## 1. Introduction

Spinal cord injury (SCI) is a severe traumatic disease that often leads to severe and permanent paralysis and carries a heavy burden for individuals, families, and society [1]. SCI occurs in a population of approximately 250,000 to 500,000 worldwide each year. In addition, according to the World Health Organization (WHO), the costs of healthcare, in both time and money, are expected to increase in the coming years [2,3]. When SCI develops, patients’ normal sensory, motor, or autonomic function is greatly affected. They often have numerous multisystem complications, such as motor deficits, gastric dysmotility, cardiac arrest, and bladder dysfunctions [4,5,6]. Based on many early primary injuries, progressive secondary injuries can exacerbate the clinical situation [7]. SCI is primarily caused by spinal misalignment and damage, resulting in immediate neuronal cell death, rupture of blood vessels, and damage to the blood–spinal-cord barrier (BSCB). In addition, the wound microenvironment triggers neuronal cell death, inflammatory responses, and vascular changes. Problems like these cause additional nerve damage and dysfunction, further prolonging SCI [8,9]. Due to the complex pathophysiology of SCI, there still remains no effective, definitive treatment or functional recovery strategy for SCI. Exosomes and microRNAs secreted from stem cells can freely pass through the blood–spinal-cord barrier. Once they reach the spinal cord, they may play a role in promoting nervous system repair, including reducing neuronal cell death, promoting vascular remodeling and neurogenesis, reducing neuroinflammation, and promoting axonal remodeling. Herein, we describe the therapeutic effect of stem cells that occurs via paracrine signaling and explain the importance of exosomes as mediators in SCI (Figure 1).

## 2. Epidemiology of SCI

SCI refers to the most common and destructive injuries in spinal surgery [10] and often leads to paralysis accompanied by a significant decline in the quality of life of the individuals and their families [1]. SCI is one of the most serious diseases because it can disrupt the connections between the brain and peripheral organs, creating a complex pathophysiology [11]. In addition, SCI patients often experience motor deficits and bladder dysfunctions, which lead to the activation of neuroinflammation, microvascular disruption, formation of glial scars, and upregulation of inhibitor factors to cell survival, and even death [12,13,14]. The SCI can be caused by physical trauma, including traffic accidents (38.6%), falls (32.2%), gunshot wounds (14.0%), sports-related injuries (7.8%), and surgical procedures (4.2%) [15]. Paraplegia induced by transient or permanent spinal cord ischemia is one of the symptoms with a high incidence. Specifically, when a cardiological surgery is performed to block and re-perfuse the aorta, such as an endovascular thoracoabdominal aortic aneurysm, the patient could become paralyzed, accidentally [16].

## 3. Pathophysiology of SCI

SCI is characterized by the rapid triggering of secondary damage after primary mechanical damage [1]. The primary injury is the direct compression of nerve elements resulting from the initial mechanical trauma, which damages both the central and peripheral nervous systems. At this time, blood vessels are damaged, and axons and neuron membranes are destroyed. After this, microhemorrhage develops in the central gray matter and spreads throughout the downstream spinal cord area, and the spinal cord swells to occupy the entire spinal canal at the site of the injury. Ischemia, the release of toxic chemicals resulting from the disruption of the neuronal membrane and electrolyte transport, causes secondary damage that substantially compounds the initial mechanical damage by killing or damaging other cells. Over time, astrocyte scarring and spinal cord voids form, which inhibit nerve axon regeneration and result in motor and sensory dysfunction below the damaged plane, even causing paralysis [1,17]. The secondary damage initiated by a primary injury can lead to post-injury hypoperfusion in the gray matter extending into the surrounding white matter. This hypoperfusion contributes to spinal shock by slowing or completely blocking the propagation of action potentials along axons [18,19]. As a result, ischemia, hypoxia, inflammatory reactions, neuronal apoptosis, and death of oligodendrocytes can be observed in the damaged area [1]. Damage to the spinal cord in this sequence can cause paralysis in parts of the body.

SCI initiates secondary injuries including ischemia, apoptosis, and inflammatory responses. The inflammation after SCI increases two types of macrophages: M1 and M2. Characterizing the M1 and M2 phenotypes can offer insights into how macrophages function in neuropathological conditions [20]. M1 macrophages are associated with T-helper 1 cytokines, such as interleukin (IL)-12 and tumor necrosis factor (TNF)-α, and M2 macrophages are related to T-helper 2 cytokines, such as IL-4 and IL-10 [21]. M2 macrophages, which are called activated macrophages, have a neuroprotective effect by affecting cellular integrity and scaffolding after nerve injury [22,23,24].

Neuronal cell death and inflammation after SCI occur concurrently through various pathways. Neuronal cell death after SCI results in an increase in the cell surface signaling pathways such as death receptor pathways; these receptors are members of the TNF receptor family and include TNF receptor 1, Fas, Fas ligand, p75, and death receptor 3 (DR3) [3]. In SCI, the Fas and p75 receptors are expressed on the surface of cells in the spinal cord including oligodendrocytes, astrocytes, and microglia, and they can recruit and activate the expression of caspase family members such as caspase-8 and caspase-10 [3]. A wide range of astrocyte reactions are initiated, and spinal membrane cells invade the damaged area to form a new glial membrane between these cells and the surviving astrocytes. As a result, the formation of scars occurs within astrocytes in the spinal cord [3,25]. In addition, the blood–brain barrier (BBB) is disrupted and the inflammatory response is increased, activating local immune cells and increasing vascular permeability [3,26,27]. T-cell numbers, microglia activation, and peripheral macrophage infiltration gradually increase in the injured center after SCI [3,28]. Microglia activation occurs 3–7 days after injury, and monocyte infiltration and macrophage activation occur within 7 days after injury [3,29].

Despite increased inflammatory events or apoptotic cell death, many signal transduction pathways can facilitate self-repair after SCI. The mitogen-activated protein kinase (MAPK) pathway is one of the important pathways regulating cell proliferation and apoptosis to activate inflammatory cytokines and to release oxidative stress products from injured cells [3,30]. The Janus kinase/signal transducer and activator of transcription (JAK/STAT) pathway is involved in various physiological processes including immune response, hematopoiesis (blood cell formation), cell proliferation, and tissue repair. Another pathway is the mammalian target protein of rapamycin (mTOR) signal transduction pathway, a protein kinase that plays a central role in regulating cellular processes related to growth, metabolism, proliferation, and survival. The JAK/STAT pathway transfers signals from the cell surface to the nucleus and regulates normal cell cycles and immune inflammatory reactions [3,31,32]. The mTOR signaling pathway regulates cell growth, differentiation, and apoptosis [3,33,34]. The mechanisms of injury and repair in SCI are unknown; however, the pathways may provide us with new ideas for treatment in the central nervous system (CNS).

## 4. Current Treatments of SCI

Currently, SCI treatment including both cellular and non-cell therapy is still the biggest challenge for researchers. Here are some common treatments and approaches for SCI. Immediate spinal cord treatment is a method to prevent aggravation of the primary injury and may require immobilization, traction, and sometimes surgery to align and stabilize the patient’s spine to prevent further damage in the acute phase of SCI [35]. Medications are used to manage a variety of symptoms associated with SCI, including pain, muscle spasms, and bladder dysfunction [36,37]. Common medications include pain relievers, muscle relaxants, and antispasmodics. Physical therapy is essential to maintaining and improving strength, flexibility, and overall mobility. It helps prevent muscle atrophy and joint contracture [38]. Functional electrical stimulation (FES) therapy uses electric current to stimulate nerves and muscles to help people with paralysis improve muscle function and potentially regain control of their limbs [39]. As for experimental treatments, researchers continue to explore new approaches to treating spinal cord injuries, including stem cell therapy, nerve regeneration technology, and neural prostheses. These experimental treatments aim to promote nerve regrowth and restore lost function [40]. Currently, there is no known cure for SCI, and treatment options focus on managing symptoms, preventing complications, and facilitating functional recovery where possible.

## 5. Mesenchymal Stem Cells (MSCs)

Stem cells are broadly divided into three types: embryonic stem cells, adult stem cells, and induced pluripotent stem cells; each type has both advantages and disadvantages. Mesenchymal stem cells (MSCs), which are one type of adult stem cell, are easy to obtain and have no ethical restrictions; they are also more easily isolated and subject to experimental usage than other types of stem cells, which are more immunogenic [41,42,43,44]. With the development of stem cell technology, MSCs have the potential to be multipotent and can be induced into many kinds of cells, such as osteoblasts (bone cells), chondrocytes (cartilage cells), myocytes (muscle cells), and adipocytes [1,45]. The use of MSCs is the most promising strategy in regenerative medicine. MSCs have overwhelming evidence for their regenerative effects in preclinical studies, and studies support their ease of isolation and preservation, rapid proliferation, limited ethical concerns, and, very importantly, their safety profile [46]. 

MSCs are a kind of pluripotent stem cells with a self-renewal ability, and they show good application prospects because they are ideal donor cells for transplantations [1,47]. MSCs exert their therapeutic effects through various paracrine mechanisms, including anti-inflammatory, immunomodulatory, neuroprotective, and stimulatory angiogenic mechanisms [48]. Through this mechanism, various tissue sources such as bone marrow mesenchymal stem cells, umbilical cord mesenchymal stem cells, adipose-derived mesenchymal stem cells, and neural stem cells can be obtained. Cells, neural progenitor cells, embryonic stem cells, induced pluripotent stem cells, and extracellular vesicles have been studied [49]. Beginning with bone marrow transplantation 60 years ago, the journey of stem cell therapy has evolved over the years and has become a new treatment in regenerative medicine to treat numerous incurable diseases [50]. However, these cells’ neural differentiation and nerve cell replacement lack a systematic description of the effects of combination therapy [51]. In addition, MSCs also secrete exosomes and biomolecules during differentiation or the pluripotent phase. 

## 6. MSCs in SCI

MSCs are known to have the ability to differentiate into neurons and express neural markers through a specific process [52,53]. MSCs secrete various factors that promote the survival of damaged neurons and oligodendrocytes and promote angiogenesis [54,55,56]. Several studies have already demonstrated that MSCs facilitate cellular survival through secreting neurotrophic factors, neuroprotective cytokines, and anti-inflammatory molecules to reduce the production of stress-associated proteins, reactive oxygen species, and proinflammatory cytokines [13,57,58]. Subsequently, factors such as glial-cell-derived neurotrophic factor (GDNF), brain-derived neurotrophic factor (BDNF), and nerve growth factor (NGF) are secreted from MSCs. They play an essential role in nerve regeneration by promoting the survival of neurons, stimulating axonal growth, guiding axonal pathfinding, and supporting the formation of functional connections within the nervous system [59]. MSCs have also been reported to exert their therapeutic effect via direct cell fusion, mitochondrial transfer, and the production of microvesicles [60]. Substantial clinical trials of MSC therapy for CNS injury have fallen short of expectations, despite promising preclinical results for treatment. Many factors influence the clinical outcome of MSC therapy. 

To date, transplantation of MSCs is a hot spot for the treatment of SCI, but many problems and risks have not been resolved. Clinical studies have reported low-grade fever, gastrointestinal dysfunction, headache, and urinary tract infection in only a minority of patients after MSC transplantation [61]. In some studies, MSC injection near the lesion site improved the condition of most patients, but only 1/14 showed improvement when administered intravenously [62]. Given these results, it presents a therapeutic challenge because the route of administration, injection site, injection time, and cell carrier material can affect the retention time, viability, and migration ability of MSCs [63].

MSC therapy after SCI may improve the functions of motor, sensory, and autonomic nerves [10,57,58,64,65]; however, several studies have demonstrated that stem cells have lower survival rates in tissues and that there are some risks such as cell de-differentiation, immune rejection, and malignant tumor formation after transplantation [66,67,68,69]. A large percentage of MSCs are taken up by the lungs and liver during circulation, while <1% of transplanted cells migrate to injured tissue [1,70]. In addition, many studies investigated the poor survival and low grafting potential of MSCs in damage areas and found limiting MSC effectiveness in tissue repair [71,72,73,74,75].

Currently, there are many different types of cell transplantation strategies used in therapeutic studies of SCI, such as neural stem cells, MSCs, and ES cells. Many researchers have studied the mechanism of stem cell therapy for SCI to relate their paracrine effects [76,77,78,79,80]. Following the studies, exosomes have been recognized as paracrine factors released by the cells. They also have a neuroprotective, angiogenic, and immunomodulatory effect and easily cross the blood–spinal-cord barrier to promote axonal regeneration [19,76].

## 7. Exosomes and Exosomal microRNAs (miRNAs)

The secretion of nano-sized lipid bilayered extracellular vesicles is a universal cellular process occurring from simple organisms to complex multicellular organisms. The term extracellular vesicles broadly contains several types of vesicles, such as exosomes, microvesicles, and apoptotic bodies. Extracellular vesicles, which are expressed in plants, mammals, bacteria, and others consist of a variety of proteins, enzymes, transcription factors, lipids, extracellular matrix proteins, receptors, and nucleic acids [81]. Recent progress in this area has revealed that extracellular vesicles play multifaceted pathophysiological functions by delivering complex messages between cells and organisms, suggesting that they play diverse roles in intracellular and interkingdom communication. Many researchers have hypothesized that the repair mechanisms of stem cells lie in the homing and replacement of damaged cells; however, the extracellular vesicles secreted by transplanted cells may be more important for repair [1,82]. MSCs produce and release a wide range of bioactive molecules called secretomes. As a result of protein analysis of secretomes, they have been found to contain nutritional factors and cytokines such as growth factors, immunomodulators, and antioxidants [83]. As biocompatible carriers, they have many advantages such as innate stability, low immunogenicity, target tissue affinity, and excellent ability to penetrate cell membranes. Therefore, the paracrine factors of MSCs have various functions such as anti-inflammation, anti-apoptosis, regulation of the extracellular matrix, and neuroprotection through protective actions against fibrosis, apoptosis, and oxidative damage [84].

Exosomes are small, single-membrane secretory organelles, measuring approximately 30–200 nm in diameter; they have the same topology as cells and are rich in selected proteins, lipids, nucleic acids, and glycoconjugates [85]. Exosomes have been detected in almost all body fluids, including urine, blood, serum, cerebrospinal fluid, saliva, and lymph [81]. Exosomes can invaginate to the cellular membrane in the first phase of exosome biogenesis and form early-sorting endosomes (ESEs) [3,86,87]. Last-sorting endosomes can develop from ESEs with an endosomal complex necessary for transport (ESCRT) proteins to generate within multivesicular bodies (MVBs) [3,88,89,90].

Exosomes are composed of various molecules, such as proteins (including receptors, adhesion molecules, tetraspanin proteins, and enzymes), lipids (including cholesterol, sphingolipids, and ceramide), and nucleic acids (including DNAs, mRNAs, and miRNAs) within and on the surface that have been identified by electron microscopy and proteomic techniques [3,85,91]. They also contain heat-shock proteins (including HSP70 and HSP90), which are implicated in antigen presentation [92]. In addition, tetraspanin proteins, including CD9, CD63, and CD81, are found in most cell types and recognized as exosome marker proteins [93].

Exosomes are released by donor cells into the extracellular environment, including intracellular communication between a parent cell and surrounding cells. Many kinds of cells can produce exosomes, which bind to the target cell surface to transmit biological information and related genetic proteins to the target cells and then finally regulate specific biological functions of cells [81,94,95]. The clinical importance of exosomes has been established in their use as alternatives to liposome-mediated drug delivery. In addition, exosomes are a promising biological gene delivery system for their microRNA and mRNAs [81]. However, there are still many aspects of exosomes that are not fully characterized. The potential of exosomes as delivery systems has diverse development in gene or drug deliveries, disease diagnostics, and biomarker-driven therapies. Exosomes can successfully deliver drugs and genes to facilitate optimal medication delivery by (1) encapsulating a sufficient number of drugs, (2) providing a consistent size, shape, and therapeutic agent throughout the blood circulation, and (3) being unharmful, immune-suppressive, and non-immunogenic [3,96,97]. Exosomes can remodel the extracellular matrix and deliver signaling molecules to other cells. This function plays an important role in various aspects of human health and disease. Due to these characteristics, exosomes are being developed as therapeutic agents for various diseases to diminish tissue damage and improve their function [1,92,98,99]. In addition, exosomes have the unique feature of readily crossing the BBB and delivering molecules to the CNS as therapeutic delivery vehicles to treat neurological disorders [4,100]. Currently, it is important to determine how to successfully transfect exosome-based drug products into cells.

There are three types of applications: incubation, electroporation, and sonication [101,102]. Incubation is the simplest method [103], and there are two types of studies; one is curcumin-loaded exosomes in LPS-induced septic shock [104], and the other is paclitaxel-loaded MSC-derived exosomes for antitumor impact [105]. The incubation method has the disadvantage of relatively low loading efficiency, so it is combined with other methods to improve the loading of small molecule drugs [106]. A method of electroporation is the fastest and can penetrate the cell membrane layer using high-voltage pulses. A major drawback of electroporation is the formation of nucleic acid and exosome aggregates during encapsulation, which will affect the function of the nucleic acid [107,108,109]. Tian et al. reported that doxycycline-loaded exosomes administered by electroporation targeted tumor tissues, resulting in tumor suppression with no damage to normal cells [110]. To load medicines, proteins, and nanoparticles, sonication can be used and applied to additional mechanical shear strain [111,112]. Although sonication is an important technique for loading drugs into exosomes, it is harmful to the bilayer membrane.

Exosomal microRNAs (miRNAs) have critical functions to silence post-transcriptional mRNA expression that is involved in cell proliferation, cell differentiation, immunomodulation, and angiogenesis [113,114]. miRNAs are a class of small noncoding RNAs regulating the expression of genes by binding the 3′-UTR of their target mRNAs negatively, which results in gene silencing [115,116]. Exosomal miRNAs are important components of exosomal functional substances and are thought to play an important role in the process of reducing neuronal cell death induced by exosomes. Processing exosomal miRNAs in SCI is recognized as a promising therapeutic approach. miRNAs can serve as diagnostic biomarkers and are emerging as novel therapeutic targets for CNS injuries [117]. To date, several studies have shown that miRNAs transported by exosomes in various cells exert significant protective effects in SCI [3]. For this reason, many studies are investigating the positive effects of miRNAs and exosomes on SCI.

## 8. Effects of Exosomes in SCI

After SCI, a large number of neuronal cells are damaged following axonal loss, ischemia, inflammation, and apoptosis [118]. Neuronal cell death has been reported to occur only in the early stages of SCI, which involves blood vessel rupture, resulting in bleeding and hypoxia [3,119]. Therefore, many studies have shown that using exosomes for treatment in the early stages after SCI can successfully attenuate the apoptosis of neurons [1,5,13,120,121]. There are several types of therapy SCI using exosomes that act by attenuating neuronal apoptosis, promoting neurogenesis, and improving axonal remodeling. For this, in the laboratory, exosomes are isolated using various methods for various purposes and applications. For example, it can be isolated using ultracentrifugation techniques [122,123,124], size-based isolation techniques [125,126], polymer precipitation [127,128], and immunoaffinity capture techniques [129,130]. Although various methods have been developed for the isolation and purification of exosomes, a combination of different isolation methods may be better than the isolation effect of a single method. Therefore, in order to obtain ideal exosomes by improving isolation efficiency and enrichment, many research teams have combined different methods of isolating and purifying exosomes to improve yield and purity [131,132].

Apoptosis, which is known as programmed cell death, can be attenuated by intervening with certain factors, such as exosomes, and reducing the ATP-driven process of cell death [133,134]. Apoptotic cell death occurs and contributes to many diseases, such as cancer, restenosis of tissues, stroke, heart failure, neurodegeneration, and AIDS, with cell accumulation and loss [134]. There are two major pathways for caspase activation: the extrinsic pathway, which is induced by the TNF family of cytokine receptors, such as TNFR1 and Fas, and the intrinsic pathway, which is induced by cytochrome c and elevations in the pro-apoptotic Bcl-2 family proteins such as Bax [134].

Exosomes and exosomal miRNAs enhance recovery from SCI by attenuating neuronal cell death. Cell apoptosis can be attenuated under the intervention of certain factors such as exosomes [133]. In preclinical experiments, MSC exosomes have been shown to increase Bcl-2 expression and decrease Bax levels after systemic administration in an SCI mouse or rat model as well as to promote functional improvement and reduce disrupted endothelial cells at the contusion site [1,120]. Another preclinical experiment demonstrated that MSC exosomes could effectively activate the Wnt/β-catenin signaling pathway with anti-apoptotic effects [121]. Other groups have also reported that MSC exosomes improved the expression of autophagy-related proteins, such as LC3IIB and Beclin-1, and induced autophagosome formation [5,13]. Liu et al. found that bone-MSC-derived exosomes had the potential to reduce lesion size, neuronal apoptosis, and glial scar formation and induced blood vessel density and axonal regeneration in SCI rats by suppressing the activation of astrocytes [13].

When SCI occurs, the injured spinal column is hypoxic to reduce endogenous neural tissue repair and tissue regeneration with abnormal angiogenesis. Vascular endothelial cells show increased uptake of exosomes to activate protein kinase A (PKA) signaling and promote vascular endothelial growth factor (VEGF) expression as a component of the blood vessel wall [135]. Cao et al. found that exosomes derived from human urine stem cells stimulated angiogenesis through the PI3K/AKT signaling pathways to enable SCI recovery in the damaged area [136]. In addition, MSC exosomes have been shown to enhance angiogenesis and axon regeneration by reducing PTEN expression in the damaged spinal cord region and to reduce microglia and astrocyte proliferation in SCI rats [137]. Exosomes have also been confirmed to promote axon regeneration by regulating the NF-κB p65 signaling pathway in pericytes [138].

Exosomes are known to contain various proteins including growth factors, such as NGF, which regulates neuronal survival and the release of neurotransmitters and facilitates the plasticity of axons in the adult central and peripheral nervous systems [4,139]. One group reported that exosomes derived from NGF-overexpressed MSCs promoted SCI repair in a mouse model, leading to the regeneration of neuronal axons and differentiation into neurons [4].

## 9. Effects of miRNAs in SCI

miRNAs are 21–23 nucleotides in length and are endogenous nonprotein-coding RNA molecules. Many researchers have predicted that the administration of miRNAs could easily reach a target region that has experienced neuronal death or a damaged organ [3,16,85,140,141,142,143]. Numerous studies have shown that miRNAs have important roles in some autoimmune diseases and are essential in the development and homeostasis of the immune system [144,145,146]. Recently, miRNAs have been identified as potential new targets for SCI treatment [147,148,149,150]. Accumulating evidence indicates that exosomes with a bilayer membrane structure can be used as valuable carriers for targeting miRNAs at SCI sites. In addition, exosomes can cross the BBB to enhance the therapeutic effect of miRNAs [151]. MSCs are pre-transfected with specific miRNA plasmids to secrete exosomes containing high levels of specific miRNAs [142]. Extensive studies have shown that exosomes from MSCs carrying miRNAs exhibit efficient repair effects in SCI.

Lin’s group reported that increased miR-126 could promote angiogenesis, inhibit inflammation, and improve functional recovery after SCI [148]. However, there remains a limitation regarding the delivery of miRNAs into the damaged tissue. Similar to siRNAs, miRNAs are highly unstable in the local extracellular environment and are delivered to the target tissues by an effective carrier system. In this regard, there are two delivery systems: viral and non-viral systems. The viral system is the most efficient delivery system but has toxic side effects, and the non-viral system contains physical and chemical components that are safe for clinical application [115,152,153,154]. Therefore, exosome-derived miR-126-treated MSCs were used as a non-viral delivery system and were found to promote angiogenesis and neurogenesis and attenuate apoptosis after SCI in a rat model [115]. Similarly, there are a few studies regarding exosomes of miRNA-modified MSCs as a therapeutic delivery vehicle to the CNS. Another group found that exosome-derived miR-133b-MSCs contributed to neurite remodeling and improved functional recovery poststroke [155]. In addition, miR-146b-transferred exosomes were found to have the potential to inhibit tumor growth in brain tumor models [156]. Zhao et al. demonstrated that exosomes from miR-25-overexpressed-MSCs exerted neuroprotective effects in an ischemic SCI model [16]. miR-25-enriched exosomes were also shown to inhibit NADPH oxidase 4 expression and increase superoxide dismutase activity and exhibited lower levels of IL-1β and TNF-α.

miR-19b and miR-21 exosomes have been found to enhance neuronal cell viability and inhibit neuronal cell death by inhibiting PTEN/PDCD4 expression. These two miRNAs are found in bone-marrow-derived MSCs, and they inhibit neuronal cell death and promote neuronal differentiation in SCI. When the number of exosomes increases, phosphatase and tensin homolog (PTENC) is inhibited preventing neuronal cell death and promoting neuronal differentiation [157,158]. BMSC-derived exosomes have been shown to have neuroprotective effects in the ischemic SCI. This effect may be due to the pre-transfection of BMSCs to secrete exosomes with high expression of miR-25, thus indicating that miR-25 enhances neuroprotection [16]. Exosomes secreted from miR-29b-rich MSCs and human neuroepithelial stem cells can exert therapeutic effects on SCI by downregulating PTEN/caspase-3 expression and subsequently inhibiting neuronal apoptosis [159,160]. Injecting exosomes secreted from BMSCs modified with miR-29b into a mouse model showed that these exosomes not only promote nerve regeneration but also accelerate motor function recovery in mice with SCI and reduce pathological damage to spinal cord tissue. This mechanism may also be related to the regulation of the expression of nerve-regeneration-related proteins such as NF200, GAP-43, and GFAP [161].

There are several studies of BMSC-derived exosomal miR-124-3p, which has been shown to attenuate neurological damage, such as in Parkinson’s disease and spinal cord ischemia-reperfusion injury [141,161]. miR-124-3p and miR-125 are responsible for regulating M2 macrophages. M2 macrophages are key effector cells of the inflammatory response to SCI, and the repair of SCI is based on macrophage activation. miR-125 promotes M2 macrophage polarization and negatively regulates IRF5 to ameliorate SCI. In addition, miR-124-3p promotes M2 macrophage polarization and negatively regulates Ern1 to ameliorate SCI [141,143]. miR-124-3p has also been found to suppress A1 astrocytes by inhibiting the activation of M1 microglia and microglia-induced neuroinflammatory responses through the MYH9/PI3K/AKT/NF-κB signaling pathway [162]. Exosomes derived from MSCs transformed with miR-126 have been shown to reduce neuronal death and promote functional regeneration after SCI. BMSC-exosome-derived miR-126 promotes angiogenic migration by repressing the expression of SPRED1 and PIK3R2 to promote SCI recovery [115]. miR-216a-5p has been found to affect the activation of NF-κB signaling by regulating TLR4. Exosomal miR-216a-5p also transforms microglia from an M1 pro-inflammatory phenotype to an M2 anti-inflammatory phenotype, increasing its therapeutic potential by inhibiting TLR4/NF-κB and activating the PI3K/Akt signaling pathway [163,164]. Exosomes secreted from BMSCs containing miR-145-5p improved functional recovery and reduced histopathological damage and inflammation in SCI mice. Exosomes promoted miR-145-5p expression in spinal cord tissue, which specifically targeted TLR4 and inhibited TLR4/NF-κB pathway activation in SCI rats [165]. MSC-derived miR-26a exosomes increased phosphorylation of PI3K, AKT, and mTOR proteins, promoting neurofilament production and nerve regeneration in neurons. miR-26a exosomes were injected into the tail vein of SCI rats to promote functional recovery in rats and induce neuronal and axonal regeneration by targeting the PTEN and mTOR pathways [166].

miR-199a-3p/145-5p, which is highly expressed in exosomes secreted from human umbilical cord mesenchymal stem cells, showed an anti-apoptosis effect in vivo. These exosomes could be an effective therapeutic strategy in neuronal injury by affecting TrkA ubiquitination and promoting the NGF/TrkA signaling pathway [167].

NSC-derived miRNA-124, which is involved in nerve regeneration, ameliorates nerve loss and reduces astrocytes in mice with SCI while increasing neurofilament-200 (NF-200) expression. It has been shown that the expression of nuclear-enriched abundant transcript 1 (Neat1) induces neuron-specific differentiation of neural stem cells by miR-124. It can also activate Wnt/β-catenin signaling to promote neuronal differentiation and migration ability [168]. NSC-derived MiR-615, which is involved in neuronal survival and axonal regeneration, inhibits LINGO-1 by directly targeting LRR and Ig domain-containing NOGO receptor-interacting protein 1 (LINGO-1), a potent negative regulator of neuronal survival and axonal regeneration, and may contribute to neuronal differentiation through the LINGO-1/RhoA or EGFR signaling pathways. Intrathecal administration of miR-615 to SCI rats suppressed LINGO-1, increased neuronal survival, enhanced axon extension and myelination, and enhanced motor function [169].

When exosomes encapsulated with miR-133b were injected into SCI rats, STAT3, ERK1/2, and CREB were found to be activated, damaged neurons were shown to be protected, and the recovery of hind limb motor function was demonstrated to be improved. RhoA is a direct target of miR-133b found in exosomes of adipose-derived stem cells. miR-133b protects neurons from apoptosis by downregulating RhoA and regulating Rho-associated kinase (ROCK) to promote ERK phosphorylation. In addition, miR-133b promotes axonal regeneration after SCI by promoting CREB and STAT3 phosphorylation [170]. Neural stem cell-derived miR-219a-2-3p has also been found to inhibit inflammation by downregulating the YY1 gene and inhibiting NF-κB [171]. miR-388-5p has been found to downregulate the expression of its target, cannabinoid receptor 1 gene (Cnr1), which increases cAMP accumulation through miR-338-5p, which activates Rap1. In turn, this activates the PI3K/AKT pathway. This pathway inhibits apoptosis and enhances neuronal survival [172].

Another exosomal miRNA with neuroprotective effects is miR-544. Rat BMSCs were transfected with miR-544 mimics to obtain exosomes highly expressing miR-544, and these exosomes were intravenously injected into a rat model of SCI [173]. The results of this study showed that miR-544 accelerated the recovery of neuronal function after SCI. In addition, overexpression of miR-544 in BMSC exosomes ameliorated histological defects and neuronal loss due to SCI [163].

The abovementioned study results demonstrate that exosome-mediated miRNA transport is a new treatment method for SCI because exosomes increase neuron activity through miRNA transport and promote functional recovery by attenuating apoptosis at an early stage during SCI. Table 1 shows the various mechanisms by which miRNAs exert a role after SCI. In addition, Table 2 shows the summary of the function of miRNAs. miRNAs can be observed to modulate processes such as neuroinflammation and apoptosis, can exert neuro-regenerative effects by targeting various molecular mechanisms, and can help recovery from SCI.

## 10. Conclusions

Exosomes, as key secretomes, have emerged as pivotal players in understanding and potentially treating SCI. This manuscript explored their multifaceted roles, from serving as transporters that can pass through the BSCB to their role in conveying miRNAs that could be instrumental in SCI treatment. The detailed analysis of various miRNAs and their mechanisms has unveiled novel pathways for intervention, highlighting the therapeutic potential of MSC-derived exosomes. Furthermore, the innovative methods discussed for delivering these miRNAs offer a promising avenue for targeted treatment. However, the translation of these findings into clinical applications requires further research, addressing challenges in delivery, specificity, and safety. Future studies should continue to explore the complex interactions of exosomes and miRNAs in SCI, paving the way for personalized and effective treatment strategies. The insights gained from this research contribute to the growing body of knowledge on SCI and represent a significant step toward innovative therapeutic solutions.

## Figures and Tables

**Figure 1 ijms-24-13849-f001:**
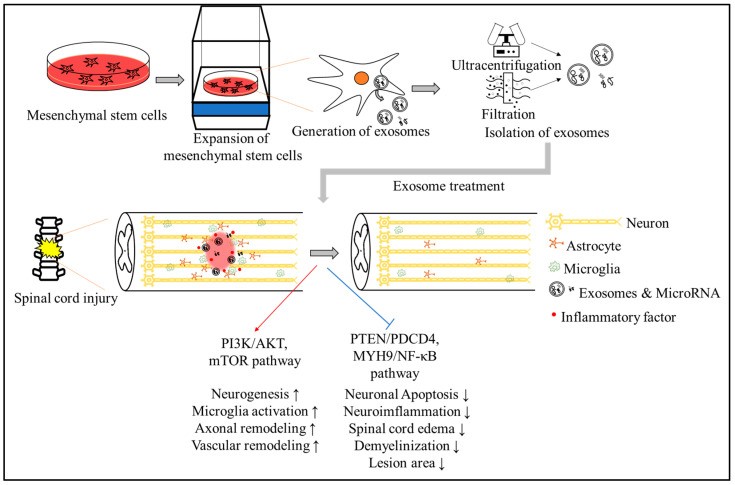
The schematic figure of effects of exosomes and microRNA in SCI. Researchers can try to repair the nervous system following SCI after stem cell-derived exosome and miRNA transplantations. The upward arrows (↑) denote stimulation, and downward arrows (↓) denote reduction.

**Table 1 ijms-24-13849-t001:** Studies on the treatment of SCI with miRNA derived from MSCs.

miRNA	Model	Mechanism	Ref.
miR-126	Rat, Contusion model, Intrathecal injection	VEGF, SPRED1, PIK3R2	[148]
	Rat, Contusion model, Intraperitoneal injection	ERK, AKT	[115]
miR-133b	Rat, Hemorrhage model, Intravenous injection	RhoA, ERK1/2, CREB	[155]
	Rat, Compression model, Intravenous injection	ERK1/2, STAT3, CRE	[170]
miR-146b	Rat, Tumor implantation model, Intratumoral injection	EGFR	[156]
miR-25	Rat, Ischemia model, Intrathecal injection	NOX2, NOX4	[16]
miR-19b	Rat, Contusion model, Intravenous injection	PTEN	[158]
miR-21	Rat, Contusion model, Intravenous injection	PTEN, PDCD4	[157]
miR-29b	Rat, Contusion model, Intravenous injection	NF200, GAP-43, GFAP	[159]
	Rat, Contusion model, Intravenous injection	PTEN, Caspase-3	[160]
miR-124-3p	Rat, Ischemia model, Intravenous injection	Ern1, Arg1, Ym1, Fizz	[141]
	Mouse, Contusion model, Intravenous injection	MYH9, PI3K, AKT, NF-κB	[161]
miR-125	Rat, Contusion model, Intrathecal injection	IRF5, Arg1, Ym1, Fizz	[143]
miR-145-5p	Rat, Transection model, Intravenous injection	TLR, NF-κB	[165]
miR-26a	Rat, Contusion model, Intravenous injection	PTEN, mTOR	[166]
miR-199a-3p	Rat, Contusion model, Intravenous injection	NGF/TrkA	[167]
MiR-124	Mouse, Transection model, Intrathecal injection	Neat1, Wnt/β-catenin	[168]
miR-615	Rat, Transection model, Subdural injection	LINGO-1, RhoA, EGFR	[169]
miR-219-a-2-3p	Rat, Contusion model, Intravenous injection	YY1, NF-κB	[171]
miR-216-5p	Mouse, Contusion model, Intravenous injection	TLR4/NF-κB/PI3K/AKT	[163]
miR-388-5p	Rat, Contusion model, Intravenous injection	cAMP, PI3K/Akt	[172]
miR-544	Rat, Contusion model, Intravenous injection	IL-1α, TNF-α, IL-17B, IL-36β	[173]

**Table 2 ijms-24-13849-t002:** Summary of the effects of miRNAs in SCI.

Effect/Function	miRNAs	Refs.
Neurogenesis	miR-126, miR-133b, miR-19b, miR-21, miR-216-5p, miR-544, miR-124, miR-615, miR-26a	[115,148,155,157,158,163,166,168,169,170,173]
Neuroprotection	miR-124-3p, miR-125	[141,143,161]
Apoptosis	miR-126, miR-133b, miR-19b, miR-21, miR-25, miR-124-3p, miR-199a-3p	[16,115,141,148,155,157,158,161,167,170]
Neuroinflammation	miR-126, miR-219-a-2-3p, miR-544, miR-145-5p	[115,148,165,171,173]

## Data Availability

Not applicable.

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
