# Peer review of "Role of Stem Cell-Derived Exosomes and microRNAs in Spinal Cord Injury"

_ijms, 2023, doi:10.3390/ijms241813849_

Round 1
Reviewer 1 Report
1. The reference is very extensive and comprehensive, demonstrating diligent research and organization to support this manuscript. Unfortunately, the review article did not provide any new information regarding application of MSC and exosomes MiRNAs in the treatment of SCI. There are several recent reviews in this area that are quite comprehensive with better illustrations. To name a few:
1. MiRNAs as Promising Translational Strategies for Neuronal Repair and Regeneration in Spinal Cord Injury Serena Silvestro, Emanuela Mazzon Cells. 2022 Jul; 11(14): 2177. Published online 2022 Jul 12. doi: 10.3390/cells11142177 PMCID: PMC9318426 2. Mesenchymal Stem Cell-Derived Exosomes: Application in Regenerative Medicine. Mangesh D. Hatde, Caitin N Surie, Zucai Suo Cells. 2021 Aug; 10(8): 1959. Published online 2021 Aug 1. doi: 10.3390/cells10081959 PMCID: PMC83934263. Exosomes combined with biomaterials in the treatment of spinal cord injuryXuanxuan Zhang, Wenwei Jiang, Yan Lu, Tiantian Mao, Yu Gu, Dingyue Ju, Chuanming DongFront Bioeng Biotechnol. 2023; 11: 1077825. Published online 2023 Mar 13. doi: 10.3389/fbioe.2023.1077825
PMCID: PMC10040754
2. Some of the paragraphs are redundant and sometimes confusing. For examples, 68-78, "As a result, ischemia, hypoxia, 68 inflammatory reactions, neuronal apoptosis, and death of oligodendrocytes can be ob-69 served in the damaged area" appeared to be part of the primary damage but then listed in the secondary damages.
Line 88-118 need to be clean up. The temporal sequence of secondary events was not well delineated or well organized.
Stem cell section 3 is poorly written. Section 4 can be more detail. Section 5 should be more about exosomes. How can exosomes synthesized in the lab, for example. Section 6 is acceptable Conclusion need to be expanded.
The whole paper need to be better organized in grammar and flow.
Author Response
Comments from Reviewer 1.
Question 1:
The reference is very extensive and comprehensive, demonstrating diligent research and organization to support this manuscript. Unfortunately, the review article did not provide any new information regarding application of MSC and exosomes MiRNAs in the treatment of SCI. There are several recent reviews in this area that are quite comprehensive with better illustrations. To name a few:
- MiRNAs as Promising Translational Strategies for Neuronal Repair and Regeneration in Spinal Cord Injury Serena Silvestro, Emanuela Mazzon Cells. 2022 Jul; 11(14): 2177. Published online 2022 Jul 12. doi: 10.3390/cells11142177 PMCID: PMC9318426
- Mesenchymal Stem Cell-Derived Exosomes: Application in Regenerative Medicine. Mangesh D. Hatde, Caitin N Surie, Zucai Suo Cells. 2021 Aug; 10(8): 1959. Published online 2021 Aug 1. doi: 10.3390/cells10081959 PMCID: PMC8393426
- Exosomes combined with biomaterials in the treatment of spinal cord injuryXuanxuan Zhang, Wenwei Jiang, Yan Lu, Tiantian Mao, Yu Gu, Dingyue Ju, Chuanming DongFront Bioeng Biotechnol. 2023; 11: 1077825. Published online 2023 Mar 13. doi: 10.3389/fbioe.2023.1077825 PMCID: PMC10040754
Answer: Thank you for your detailed review and valuable comments. We have added the references and the sentences to notice new information regarding application of MSC and exosomes miRNAs in the treatment of SCI as your recommendation. All the sentences that I changed following your comments, we have marked as blue color.
Question 2:
Some of the paragraphs are redundant and sometimes confusing. For examples, 68-78, "As a result, ischemia, hypoxia, 68 inflammatory reactions, neuronal apoptosis, and death of oligodendrocytes can be ob-69 served in the damaged area" appeared to be part of the primary damage but then listed in the secondary damages.
Answer: Thank you for your detailed review and valuable comments. As your recommendation, we have changed the sentence more concisely.
Question 3:
Line 88-118 need to be clean up. The temporal sequence of secondary events was not well delineated or well organized.
Answer: Thank you for your detailed review and valuable comments. We have changed all the part more organized following your recommendation.
Question 4:
Stem cell section 3 is poorly written. Section 4 can be more detail. Section 5 should be more about exosomes. How can exosomes synthesized in the lab, for example. Section 6 is acceptable Conclusion need to be expanded.
Answer: Thank you for your detailed review and valuable comments. We have added our hypothesis in the sections; Stem cell, MSC in SCI, and exosomes. In addition, we have added the sentences to explain how can exosomes isolated and synthesized in the lab. As your recommendation, we have added the s
Reviewer 2 Report
This manuscript delves into the complex and promising field of stem cell-derived exosomes and microRNAs (miRNAs) in the treatment of spinal cord injury (SCI). It explores the role of exosomes as secretomes that have various functions in anti-inflammation, anti-apoptosis, and neuroprotection. These extracellular vesicles are rich in proteins, lipids, nucleic acids, and specific miRNAs, which are found to have therapeutic implications for various diseases, including SCI.
Through a detailed examination, the manuscript highlights the methods of utilizing exosomes for drug and gene delivery, the applications of specific miRNAs in SCI treatment, and the potential mechanisms that miRNAs may use to facilitate repair and recovery in SCI. The conclusion emphasizes the future potential of exosomes and miRNAs in SCI treatment, underscoring their ability to pass through biological barriers and serve as transporters.
Overall, this manuscript is a valuable contribution to the field of regenerative medicine and neurology. It sheds light on the multifaceted roles of exosomes and miRNAs in SCI treatment, offering insights into potential therapeutic avenues. While the manuscript is strong in its scientific content and relevance, improvements in readability, visual representation, and language refinement could elevate its overall quality. It serves as an informative resource for researchers and clinicians interested in novel approaches to SCI treatment and provides a solid foundation for future research in this exciting area.
Below are some suggestions to enhance the quality of this manuscript:
Abstract
The abstract author has provided is well-written and succinctly summarizes the focus of the manuscript, highlighting the importance of Mesenchymal stem cells (MSCs), exosomes, and exosomal miRNAs in the context of neurological disorders. However, there are a few areas where it could be improved for clarity and coherence.
1. The introduction of the abstract mentions the short-term symptomatic and dose-dependent side effects of pharmacological treatments but doesn't specify what these side effects are. Providing a brief example or elaboration might make this point clearer.
2. The abstract could be slightly restructured to improve the flow of information. Starting with the challenge in neurological disorders, moving to the promise of MSCs, and then to the specific focus on exosomes and miRNAs might make the progression of ideas more logical.
3. While the abstract does cover the main topics, it might benefit from a more specific statement about the primary focus of the review. Is it on the mechanisms of action, therapeutic applications, challenges in the field, or something else? Making this clear will guide the reader's expectations.
Here's an example of how you could revise your abstract:
"Neurological disorders present a global health challenge, and current pharmacological treatments often lead to short-term symptomatic relief but can have dose-dependent side effects. Mesenchymal stem cells (MSCs) offer a promising alternative, with their pluripotent, anti-inflammatory, anti-apoptotic, and immunomodulatory properties. Although MSCs themselves rarely differentiate into neurons at injury sites, their secreted paracrine factors have shown therapeutic effects. Recent studies attribute the pleiotropic effects of MSCs to these factors, with exosomes playing a vital role. In addition to exosomes, various miRNAs derived from MSCs have been identified to regulate genes and mitigate neuropathological changes in neurodegenerative diseases. This review explores the burgeoning field of exosome-based therapies, focusing on the effects of exosomes and exosomal miRNAs derived from MSCs, and summarizes recent findings that illuminate their potential in treating neurological disorders. The insights gained from this review may pave the way for innovative and effective treatments for these complex conditions."
Introduction and other sections
4. Add a concise summary at the beginning to guide readers through the main topics, including SCI's impact, current treatments, and MSCs' role. This will set the context and allow readers to quickly grasp the focus of the paper.
5. Break the text into subsections with relevant subheadings like "Epidemiology of SCI," "Pathophysiology," "Current Treatments," and "MSCs in SCI." This structural organization will enhance readability and guide the reader through the manuscript.
6. Break long and complex sentences into shorter sentences for clarity, especially in detailed sections like lines 34-41. This will aid in understanding and prevent the loss of crucial details.
7. Clarify or correct ambiguous terms such as replacing "secondaries injury" with "secondary injuries" or another appropriate term (line 79). This will prevent confusion.
8. Provide more insights into how MSCs exert therapeutic effects on SCI, including detailed information on cellular interactions and growth factors involved (lines 118-154). This will provide a comprehensive understanding of the subject.
9. Expand on the challenges and limitations in MSC therapy, such as survival rates, risks, and clinical trial outcomes (lines 146-154). This critical evaluation will provide a balanced view of the topic.
10. Ensure clarity in figures by providing a clear and well-labeled Figure 1, with a detailed caption (line 44). Visual aids can significantly enhance the understanding of complex topics.
11. Use consistent terminology throughout the text, such as choosing either "neurons" or "nerve cells" (lines 63-65). Consistency aids in readability.
12. Define or briefly explain specific scientific terms and pathways, such as providing a definition for MAPK (lines 105-107). This will make the content accessible to a wider audience.
13. Summarize the current state of SCI treatment and outline future directions, including challenges and potential advancements in SCI treatment and MSC therapy.
14. Include more illustrative figures or diagrams to depict mechanisms, pathways, or summaries of key findings. These visuals could make dense information more digestible.
15. Some parts, such as the mechanisms of various miRNAs, might contain repetitive information. Consolidate these sections to avoid redundancy.
16. While heavy on technical and scientific terminology, consider providing explanations for non-specialized readers.
17. Discuss the comparative advantages, disadvantages, or contexts where applications like incubation, electroporation, and sonication might be applied.
Conclusion
The conclusion provided in the manuscript succinctly summarizes the major findings and implications of the research, specifically focusing on the role of exosomes and microRNAs (miRNAs) in the treatment of SCI. Below, I'll provide some comments and suggestions to further strengthen this conclusion.
18. While the conclusion briefly touches on the key aspects of exosomes and miRNAs, it could benefit from a more comprehensive summary of the main findings, mechanisms, and applications that were discussed in the manuscript. This would provide a more complete wrap-up of the research.
19. The conclusion should emphasize why the findings are novel or significant, especially in the context of current knowledge and treatment paradigms for SCI. This would help readers understand the unique contribution of this research.
20. It would be beneficial to elaborate on how the findings may translate into clinical applications, providing insights into how exosomes and miRNAs might be used in therapeutic strategies for SCI, and what challenges might need to be addressed.
21. Consider adding a section on future directions, outlining what further research or developments are needed to realize the potential of exosomes and miRNAs in SCI treatment. This could include potential challenges, unanswered questions, or areas where further exploration is needed.
22. It might be helpful to briefly connect back to some of the specific findings or sections of the manuscript to provide a cohesive conclusion. This could include referencing key mechanisms, therapeutic strategies, or particular miRNAs that were discussed.
Here's an example of how you could revise your conclusion:
"Exosomes, as key secretomes, have emerged as pivotal players in understanding and potentially treating spinal cord injury (SCI). This manuscript has explored their multifaceted roles, from serving as transporters that can pass through the BSCB to their role in conveying miRNAs that could be instrumental in SCI treatment. The detailed analysis of various miRNAs and their mechanisms has unveiled novel pathways for intervention, highlighting the therapeutic potential of mesenchymal stem cell (MSC)-derived exosomes. Furthermore, the innovative methods discussed for delivering these miRNAs offer a promising avenue for targeted treatment. However, the translation of these findings into clinical applications requires further research, addressing challenges in delivery, specificity, and safety. Future studies should continue to explore the complex interactions of exosomes and miRNAs in SCI, paving the way for personalized and effective treatment strategies. The insights gained from this research contribute to the growing body of knowledge on SCI and represent a significant step toward innovative therapeutic solutions."
Overall, the English in the manuscript is of good quality but could benefit from careful editing to enhance readability and flow. This may include simplifying complex sentences, ensuring consistency in terminology, and careful proofreading for grammatical accuracy. It may be beneficial to have a native English-speaking colleague or professional scientific editor review the manuscript to polish the language further.
Author Response
Comments from Reviewer 2.
This manuscript delves into the complex and promising field of stem cell-derived exosomes and microRNAs (miRNAs) in the treatment of spinal cord injury (SCI). It explores the role of exosomes as secretomes that have various functions in anti-inflammation, anti-apoptosis, and neuroprotection. These extracellular vesicles are rich in proteins, lipids, nucleic acids, and specific miRNAs, which are found to have therapeutic implications for various diseases, including SCI.
Through a detailed examination, the manuscript highlights the methods of utilizing exosomes for drug and gene delivery, the applications of specific miRNAs in SCI treatment, and the potential mechanisms that miRNAs may use to facilitate repair and recovery in SCI. The conclusion emphasizes the future potential of exosomes and miRNAs in SCI treatment, underscoring their ability to pass through biological barriers and serve as transporters.
Overall, this manuscript is a valuable contribution to the field of regenerative medicine and neurology. It sheds light on the multifaceted roles of exosomes and miRNAs in SCI treatment, offering insights into potential therapeutic avenues. While the manuscript is strong in its scientific content and relevance, improvements in readability, visual representation, and language refinement could elevate its overall quality. It serves as an informative resource for researchers and clinicians interested in novel approaches to SCI treatment and provides a solid foundation for future research in this exciting area.
Below are some suggestions to enhance the quality of this manuscript:
Question 1:
Abstract
The abstract author has provided is well-written and succinctly summarizes the focus of the manuscript, highlighting the importance of Mesenchymal stem cells (MSCs), exosomes, and exosomal miRNAs in the context of neurological disorders. However, there are a few areas where it could be improved for clarity and coherence. The introduction of the abstract mentions the short-term symptomatic and dose-dependent side effects of pharmacological treatments but doesn't specify what these side effects are. Providing a brief example or elaboration might make this point clearer.
Answer: Thank you for your detailed review and valuable comments. We have changed the abstract and added the sentences as your recommendation. All the sentences that I changed following your comments, we have marked as blue color.
Question 2:
The abstract could be slightly restructured to improve the flow of information. Starting with the challenge in neurological disorders, moving to the promise of MSCs, and then to the specific focus on exosomes and miRNAs might make the progression of ideas more logical.
Answer: Thank you for your detailed review and valuable comments. As your recommendation, we have changed and organized the abstract.
Question 3:
While the abstract does cover the main topics, it might benefit from a more specific statement about the primary focus of the review. Is it on the mechanisms of action, therapeutic applications, challenges in the field, or something else? Making this clear will guide the reader's expectations.
Here's an example of how you could revise your abstract
"Neurological disorders present a global health challenge, and current pharmacological treatments often lead to short-term symptomatic relief but can have dose-dependent side effects. Mesenchymal stem cells (MSCs) offer a promising alternative, with their pluripotent, anti-inflammatory, anti-apoptotic, and immunomodulatory properties. Although MSCs themselves rarely differentiate into neurons at injury sites, their secreted paracrine factors have shown therapeutic effects. Recent studies attribute the pleiotropic effects of MSCs to these factors, with exosomes playing a vital role. In addition to exosomes, various miRNAs derived from MSCs have been identified to regulate genes and mitigate neuropathological changes in neurodegenerative diseases. This review explores the burgeoning field of exosome-based therapies, focusing on the effects of exosomes and exosomal miRNAs derived from MSCs, and summarizes recent findings that illuminate their potential in treating neurological disorders. The insights gained from this review may pave the way for innovative and effective treatments for these complex conditions."
Answer: Thank you for your detailed review and valuable comments. In this review, we want to suggest the therapeutic effect of exosomes and miRNA exosomes from MSCs, which have a potential to rescue in spinal cord injury via diverse signaling pathways. We have added the sentences that we want to explain in this manuscript. In addition, we have changed the abstract as your recommendation.
Introduction and other sections
Question 4:
Add a concise summary at the beginning to guide readers through the main topics, including SCI's impact, current treatments, and MSCs' role. This will set the context and allow readers to quickly grasp the focus of the paper.
Answer: Thank you for your detailed review and valuable comments. As your comments, we have added the sub-titles in the manuscript.
Question 5:
Break the text into subsections with relevant subheadings like "Epidemiology of SCI," "Pathophysiology," "Current Treatments," and "MSCs in SCI." This structural organization will enhance readability and guide the reader through the manuscript.
Answer: Thank you for your detailed review and valuable comments. As your comments, we have added the sub-titles in the manuscript.
Question 6:
Break long and complex sentences into shorter sentences for clarity, especially in detailed sections like lines 34-41. This will aid in understanding and prevent the loss of crucial details.
Answer: Thank you for your detailed review and valuable comments. We have changed the sentence shorter as your recommendation.
Question 7:
Clarify or correct ambiguous terms such as replacing "secondaries injury" with "secondary injuries" or another appropriate term (line 79). This will prevent confusion.
Answer: Thank you for your detailed review and valuable comments. We have changed the sentence as your recommendation.
Question 8:
Provide more insights into how MSCs exert therapeutic effects on SCI, including detailed information on cellular interactions and growth factors involved (lines 118-154). This will provide a comprehensive understanding of the subject.
Answer: Thank you for your detailed review and valuable comments. Following your comments, we have added the sentences for detailed information.
Question 9:
Expand on the challenges and limitations in MSC therapy, such as survival rates, risks, and clinical trial outcomes (lines 146-154). This critical evaluation will provide a balanced view of the topic.
Answer: Thank you for your detailed review and valuable comments. Following your comments, we have added the sentences for detailed information.
Question 10:
Ensure clarity in figures by providing a clear and well-labeled Figure 1, with a detailed caption (line 44). Visual aids can significantly enhance the understanding of complex topics.
Answer: Thank you for your detailed review and valuable comments. We have changed the Figure 1 to clarify our hypothesis as your comments.
Question 11:
Use consistent terminology throughout the text, such as choosing either "neurons" or "nerve cells" (lines 63-65). Consistency aids in readability.
Thank you for your detailed review and valuable comments. Following your comments, we have changed the word.
Question 12:
Define or briefly explain specific scientific terms and pathways, such as providing a definition for MAPK (lines 105-107). This will make the content accessible to a wider audience.
Answer: Thank you for your detailed review and valuable comments. Following your comments, we have added the sentences for detailed information.
Question 13:
Summarize the current state of SCI treatment and outline future directions, including challenges and potential advancements in SCI treatment and MSC therapy.
Answer: Thank you for your detailed review and valuable comments. Following your comments, we have added the sentences for detailed information.
Question 14:
Include more illustrative figures or diagrams to depict mechanisms, pathways, or summaries of key findings. These visuals could make dense information more digestible.
Answer: Thank you for your detailed review and valuable comments. We have changed the Figure 1 to clarify our hypothesis as your comments.
Question 15:
Some parts, such as the mechanisms of various miRNAs, might contain repetitive information. Consolidate these sections to avoid redundancy.
Answer: Thank you for your detailed review and valuable comments. As your recommendation, we have organized the sentence more concisely.
Question 16:
While heavy on technical and scientific terminology, consider providing explanations for non-specialized readers.
Answer: Thank you for your detailed review and valuable comments. We have added the sentence for non-specialized readers to explain the technical and scientific terminology as your comments.
Question 17:
Discuss the comparative advantages, disadvantages, or contexts where applications like incubation, electroporation, and sonication might be applied.
Answer: Thank you for your detailed review and valuable comments. We have added the sentence as your recommendation.
Conclusion
The conclusion provided in the manuscript succinctly summarizes the major findings and implications of the research, specifically focusing on the role of exosomes and microRNAs (miRNAs) in the treatment of SCI. Below, I'll provide some comments and suggestions to further strengthen this conclusion.
Question 18:
While the conclusion briefly touches on the key aspects of exosomes and miRNAs, it could benefit from a more comprehensive summary of the main findings, mechanisms, and applications that were discussed in the manuscript. This would provide a more complete wrap-up of the research.
Answer: Thank you for your detailed review and valuable comments. We have added the sentence to explain our hypothesis in this manuscript.
Question 19:
The conclusion should emphasize why the findings are novel or significant, especially in the context of current knowledge and treatment paradigms for SCI. This would help readers understand the unique contribution of this research.
Answer: Thank you for your detailed review and valuable comments. We have added the sentences as your recommendation.
Question 20:
It would be beneficial to elaborate on how the findings may translate into clinical applications, providing insights into how exosomes and miRNAs might be used in therapeutic strategies for SCI, and what challenges might need to be addressed.
Answer: Thank you for your detailed review and valuable comments. We have added the sentences as your recommendation.
Question 21:
Consider adding a section on future directions, outlining what further research or developments are needed to realize the potential of exosomes and miRNAs in SCI treatment. This could include potential challenges, unanswered questions, or areas where further exploration is needed.
Answer: Thank you for your detailed review and valuable comments. We have added the sentences as your recommendation.
Question 22:
It might be helpful to briefly connect back to some of the specific findings or sections of the manuscript to provide a cohesive conclusion. This could include referencing key mechanisms, therapeutic strategies, or particular miRNAs that were discussed.
Here's an example of how you could revise your conclusion:
"Exosomes, as key secretomes, have emerged as pivotal players in understanding and potentially treating spinal cord injury (SCI). This manuscript has explored their multifaceted roles, from serving as transporters that can pass through the BSCB to their role in conveying miRNAs that could be instrumental in SCI treatment. The detailed analysis of various miRNAs and their mechanisms has unveiled novel pathways for intervention, highlighting the therapeutic potential of mesenchymal stem cell (MSC)-derived exosomes. Furthermore, the innovative methods discussed for delivering these miRNAs offer a promising avenue for targeted treatment. However, the translation of these findings into clinical applications requires further research, addressing challenges in delivery, specificity, and safety. Future studies should continue to explore the complex interactions of exosomes and miRNAs in SCI, paving the way for personalized and effective treatment strategies. The insights gained from this research contribute to the growing body of knowledge on SCI and represent a significant step toward innovative therapeutic solutions."
Answer: Thank you for your detailed review and valuable comments. We have added the sentences as your recommendation.
Question 23:
Overall, the English in the manuscript is of good quality but could benefit from careful editing to enhance readability and flow. This may include simplifying complex sentences, ensuring consistency in terminology, and careful proofreading for grammatical accuracy. It may be beneficial to have a native English-speaking colleague or professional scientific editor review the manuscript to polish the language further.
Answer: Thank you for your detailed review and valuable comments. All the sentences that I changed following your comments, we have marked as blue color. In addition, we attached the Editorial Certificate in English.

Reviewer 3 Report
The current review doesn`t address new information related to exosomes and miRNA for treated SCI. There are a considerable number of reviews in this field discussing the same topics and mechanisms.
The manuscript "The Therapeutic Effects of Stem Cell-Derived Exosomes and MicroRNAs in Spinal Cord Injury" discusses the therapeutic potential of exosomes and miRNA for SCI treatment. In fact, it is promising, especially due to the ability of these vesicles to modulate the injury microenvironment. In addition, the authors mentioned several very know mechanisms that explain the exosomes and miRNA effects such as anti-inflammatory, anti-apoptotic, and immunomodulatory proprieties. However it`s not clear, What the current review does it add to the subject area compared with other published studies? What is the relevant information here?
Major comments:
- The authors should consider changing the title. The current title suggests an experimental study with therapeutic effects evaluated in any experimental model. The authors can consider as a title suggestion: "Role of Stem Cell-Derived Exosomes and MicroRNAs in Spinal Cord Injury". Looks more appropriate for a review.
- Figure 1 illustrates the schematic effects of exosomes in spinal cord injury. However, the image suggests more of an experimental designer using the exosomes in a rat model than the effects of them. The authors should consider creating a new version emphasizing the anti-inflammatory, anti-apoptotic, and immunomodulatory or activation pathways involved, showing the author's hypothesis for exosomes and miRNA mechanisms.
- Table 1 . summarizes studies on the treatment of SCI with miRNA derived from MSCs. The authors should consider including the miRNA 145-5p and 199a-3p (recent studies published in 2021 doi: 10.1186/s13287-021-02148-5 ; doi: 10.1080/15384101.2021.1919825).
Author Response
Comments from Reviewer 3.
The current review doesn`t address new information related to exosomes and miRNA for treated SCI. There are a considerable number of reviews in this field discussing the same topics and mechanisms.
The manuscript "The Therapeutic Effects of Stem Cell-Derived Exosomes and MicroRNAs in Spinal Cord Injury" discusses the therapeutic potential of exosomes and miRNA for SCI treatment. In fact, it is promising, especially due to the ability of these vesicles to modulate the injury microenvironment. In addition, the authors mentioned several very know mechanisms that explain the exosomes and miRNA effects such as anti-inflammatory, anti-apoptotic, and immunomodulatory proprieties. However it`s not clear, What the current review does it add to the subject area compared with other published studies? What is the relevant information here?
Major comments:
Question 1:
The authors should consider changing the title. The current title suggests an experimental study with therapeutic effects evaluated in any experimental model. The authors can consider as a title suggestion: "Role of Stem Cell-Derived Exosomes and MicroRNAs in Spinal Cord Injury". Looks more appropriate for a review.
Answer: Thank you for your detailed review and valuable comments. As your recommendation, we have changed the title and marked as blue color. All the sentences that I changed following your comments, we have marked as blue color.
Question 2:
Figure 1 illustrates the schematic effects of exosomes in spinal cord injury. However, the image suggests more of an experimental designer using the exosomes in a rat model than the effects of them. The authors should consider creating a new version emphasizing the anti-inflammatory, anti-apoptotic, and immunomodulatory or activation pathways involved, showing the author's hypothesis for exosomes and miRNA mechanisms.
Answer: Thank you for your detailed review and valuable comments. We have changed Figure 1 and added the mechanisms of anti-inflammatory, anti-apoptotic, and immunomodulatory or activation pathway following your comments.
Question 3:
Table 1. summarizes studies on the treatment of SCI with miRNA derived from MSCs. The authors should consider including the miRNA 145-5p and 199a-3p (recent studies published in 2021 doi: 10.1186/s13287-021-02148-5; doi: 10.1080/15384101.2021.1919825).
Answer: Thank you for your detailed review and valuable comments. Following your recommendation, we have added the references and the sentences.

Round 2
Reviewer 2 Report
The author has addressed all of my concerns. However, I have only one minor suggestion: please list the references in Table 2.
The manuscript's language is suitable for an academic or professional audience, with appropriate use of technical vocabulary and clear expression of complex ideas. However, there are areas where grammar, syntax, word choice, and flow can be refined to enhance clarity and readability. The suggested improvements are minor and can be addressed with careful editing.
1. Some sentences could be rephrased for clarity and grammatical correctness. For example, "In addition, the costs considerable time and money for health-care will be getting increased up to the next according to the World Health Organization (WHO)" could be rephrased as "In addition, according to the World Health Organization (WHO), the costs of healthcare, in both time and money, are expected to increase in the coming years."
2. The manuscript should maintain consistency in the use of terms. For example, using either "spinal cord injury" or "SCI" consistently throughout the text.
3. Some word choices could be refined for precision and to avoid redundancy. For instance, instead of "caused by physical trauma, such as traffic accidents (38.6%), falls (32.2%), gunshot wounds (14.0%), sports-related injuries (7.8%), and surgery (4.2%)", you could say "caused by physical trauma, including traffic accidents (38.6%), falls (32.2%), gunshot wounds (14.0%), sports-related injuries (7.8%), and surgical procedures (4.2%)."
Author Response
The author has addressed all of my concerns. However, I have only one minor suggestion:
Question 1:
please list the references in Table 2.
Answer: Thank you for your detailed review and valuable comments. We have added the
references in Table 2 as your recommendation.
Comments on the Quality of English Language
The manuscript's language is suitable for an academic or professional audience, with
appropriate use of technical vocabulary and clear expression of complex ideas. However, there
are areas where grammar, syntax, word choice, and flow can be refined to enhance clarity and
readability. The suggested improvements are minor and can be addressed with careful editing.
Question 2:
Some sentences could be rephrased for clarity and grammatical correctness. For example, "In
addition, the costs considerable time and money for health-care will be getting increased up to
the next according to the World Health Organization (WHO)" could be rephrased as "In
addition, according to the World Health Organization (WHO), the costs of healthcare, in both
time and money, are expected to increase in the coming years."
Answer: Thank you for your detailed review and valuable comments. As your recommendation,
we have changed the sentence.
Question 3:
The manuscript should maintain consistency in the use of terms. For example, using either
"spinal cord injury" or "SCI" consistently throughout the text.
Answer: Thank you for your detailed review and valuable comments. We have checked all the
manuscript and written consistently SCI instead of spinal cord injury.
Question 4:Some word choices could be refined for precision and to avoid redundancy. For instance,
instead of "caused by physical trauma, such as traffic accidents (38.6%), falls (32.2%), gunshot
wounds (14.0%), sports-related injuries (7.8%), and surgery (4.2%)", you could say "caused
by physical trauma, including traffic accidents (38.6%), falls (32.2%), gunshot wounds (14.0%),
sports-related injuries (7.8%), and surgical procedures (4.2%)."
Answer: Thank you for your detailed review and valuable comments. We have changed the
sentence as your recommendation.

Reviewer 3 Report
The authors provided a new version of the manuscripts and added complementary and relevant information improving the review. Also, they changed the points recommended in the first round of the review process. I encourage the publication of the current manuscript.
Author Response
The authors provided a new version of the manuscripts and added complementary and relevant information improving the review. Also, they changed the points recommended in the first round of the review process. I encourage the publication of the current manuscript.
Answer: Thank you for your suggestion of publication.
